# Impact of Social Support on Depressive Symptoms among Postgraduates during the Coronavirus Disease 2019 Pandemic: A Moderated Mediation Analysis

**DOI:** 10.3390/ijerph20043431

**Published:** 2023-02-15

**Authors:** Peng Wan, Jinsheng Hu, Qi Li

**Affiliations:** School of Psychology, Liaoning Normal University, Dalian 116023, China

**Keywords:** graduate student, social support, depressive symptoms, positive coping, neuroticism, COVID-19

## Abstract

The coronavirus disease (COVID-19) pandemic continues to spread worldwide, and its related stressors are causing a high prevalence of mental health problems among graduate students. This has the potential for long-term effects on their mental well-being. However, few large-scale studies have been conducted on multiple risk and protective factors. Therefore, we aimed to test the impact of social support on depressive symptoms among graduate students and analyze the mediating role of positive coping and the regulatory role of neuroticism. From 1–8 October 2021, 1812 Chinese graduate students were surveyed online. We used a structural equation model to study the mediating role of positive coping in the relationship between social support and depressive symptoms and used the Hayes PROCESS macro to conduct mediating analysis. The incidence of depressive symptoms was 10.40%. These results showed that positive coping influenced the social support’s influence on depression symptoms to some extent. Moreover, neuroticism regulates the indirect relationship between social support and depressive symptoms through active coping. Further research is needed to assess the impact of various forms of social support on graduate students’ mental health and to develop strategies for maintaining their well-being, such as network mindfulness.

## 1. Introduction

The 2019 coronavirus pandemic has posed a worldwide health hazard, endangering human life and well-being and leading to severe economic and social repercussions [1]. Measures such as maintaining social distancing, abiding by a stay-at-home order, and suspending school teaching have been adopted worldwide to contain the virus. Adhering to social distancing is the most prominent way to safeguard public health during the COVID-19 pandemic. Appropriate physical distancing, self-quarantine, and contact tracing should be implemented simultaneously to impede the dissemination of the virus effectively [2]. Although these social isolation measures effectively control the spread of the disease, they have also caused some issues, with the alteration in social support being the most prominent. Social support, a form of mutual communication and connection, is generally divided into three categories: emotional, tool, and information support [3]. Social connections between people are fundamental to maintaining mental health. During the COVID-19 pandemic, governmental stay-at-home orders and personal social-distancing practices have been linked to various psychological issues [4]. Numerous studies have demonstrated the detrimental impact of the home policy and the pandemic lockdown, particularly on graduate students. The outcomes of this research include the mental and psychological issues, such as the perception of stress and symptoms of depression and anxiety [5,6], suicidal ideation [7,8,9], post-traumatic stress disorder [10], and sleep disturbances, such as insomnia [11,12]. The pandemic has caused a reliance on mobile devices and extended screen-use time [13], leading to heightened levels of loneliness and social isolation [14], and increased social avoidance and feelings of distress [15]. In addition, it is associated with a decreased quality of life and lower subjective well-being [16,17]. Without appropriate intervention, the psychological effects of the pandemic may lead to chronic mental health problems. As a result of the policy response to coronavirus disease, college student groups are living in isolation, which brings with it the uncertainty of academic and research progress [11] as well as increased academic pressure [18].

Although the above studies have explored the link between social support and mental health, the causal mechanism by which social support affects mental health remains to be analyzed. In addition, the mediating mechanisms between social support and mental health during the pandemic (i.e., how is social support related to mental health?) Moreover, regulatory mechanisms (i.e., when does the relationship work best?). However, this is also largely unknown. Answering these questions is essential to understand the mental health of graduate students better and reduce the risk of mental illness to ensure their academic and professional development during the COVID-19 pandemic.

### 1.1. Social Support and Depressive Symptoms

High prevalence of mental illness is a significant hindrance to graduate students’ life satisfaction and academic success, and high rates of mental ill-health among postgraduate researchers represent a significant barrier to life satisfaction and academic success. It is essential to pay attention to depression, a significant psychosocial outcome that may arise from the coronavirus pandemic. It is a well-established fact that the lack of social support contributes significantly to the mental health issues (including depressive symptoms) of college students [19]. The classic stress buffer model suggests that social support can create a buffer for maintaining mental health [20]. Studies conducted during the lockdown of the COVID-19 pandemic found that individuals who lack social contact [9] and experience self-isolation [21] have higher rates of depression, irritability, and loneliness. In contrast, individuals with higher levels of social support have a 63% lower risk of depressive symptoms [21]. Graduate students have been profoundly affected by the COVID-19 pandemic as they face a heightened sense of pressure and conflict between academic and life balance goals and roles, thus unprecedentedly increasing their mental burden. For example, low levels of social support and loneliness are significant risk factors for depression, anxiety, and suicide among graduate students [22]. In light of the psychological stress caused by the current wave of the novel coronavirus disease and uncertainty regarding future trends, improving social support measures can buffer the negative impact of COVID-19-related stressors [23,24]. These results suggest that social support plays a vital role in times of stress and that access to social and psychological resources can help combat psychological stressors during a pandemic.

Concurrently, the network conceptual model of social support demonstrates that social networks can affect mental health through behavioral means, including social influence and participation [25]. It has been documented that college students who live on their own or have strained relationships with their partners, peers, and acquaintances tend to have higher levels of depression and anxiety [26]. By contrast, well-connected peer networks provide individuals with beneficial positive experiences [27] and are associated with academic achievement and happiness [28,29]. Graduate students in stable relationships and living with others showed lower depressive symptoms during the coronavirus pandemic [30]. It is especially relevant to the current research that having a reliable social support system, be it through a network of friends or a strong family bond, can effectively buffer the mental impact of COVID-19 on graduate students [13]. These findings suggest that high levels of social support offer individuals the possibility of positive emotions and psychological boosts, thereby reducing their risk of psychiatric symptoms. Based on the above analysis, we proposed hypothesis H1: Social support negatively predicts depressive symptoms in graduate students.

### 1.2. Mediating Role of Positive Coping

Although many studies have found a direct relationship between social support and depressive symptoms, more research is needed on the underlying mechanisms. In addition to social support, positive coping strategies have been consistently reported to prevent mental health problems when experiencing stressors. The stress response model investigates whether an external stressor acting on an individual produces stress is related to the individual’s coping style and cognitive evaluation [31]. Coping is ‘‘a constantly changing cognitive and behavioral effort to manage specific external or internal demands appraised as taxing or exceeding the person’s resources” and includes both positive and negative coping [32]. Active and effective coping is a way to deal with stress, and some studies have found a relationship between social support and positive coping. For example, high social support was positively correlated with positive coping, and the provision of social support increased the level of individual positive coping [33,34]. In the current primary health crisis caused by the COVID-19 epidemic, mental resilience and positive coping styles are negatively correlated with depression [35]. Positive coping styles have been proven protective factors against depression, anxiety, and stress among college students, while negative coping styles aggravate psychological problems [36]. Graduate students with high mental resilience and active coping skills can effectively cope with depression, anxiety, and other issues caused by the COVID-19 pandemic [5]. These results demonstrate the benefits of healthy coping systems and engaging in rewarding activities that can alleviate students’ general negative emotions. Studies of undergraduate, graduate, and other adult groups during the pandemic found that increases in positive coping strategies and social support were significantly associated with reductions in psychological distress [32]. Even people who are abused in childhood have improved adult mental health owing to positive coping and supportive relationships [37]. This finding suggests that increased social support makes individuals more inclined to adopt positive coping styles, cope with stressful challenges, and promote mental health.

When faced with a stressor or a group of stressors, individuals are forced to consider their coping resources and choose coping responses accordingly. According to the dysfunctional hypothesis, abnormal coping may undermine the social support buffer, thus reducing mental health levels [38]. The lack of support and understanding from family members and relatives affects the individuals’ choice of coping styles [39]. During the pandemic, the negative coping styles of medical college students are positively correlated with depression [35], and negative coping strategies have a negative regulatory effect on students’ mental health [36]. In addition, other studies have found that even though college students feel a high level of social support, continuing to use dysfunctional negative coping strategies reduces their happiness [40]. These findings suggest that, when faced with stress, college students naturally seek support from family and friends and implement their coping measures to stay healthy. Based on the above discussion, we proposed hypothesis H2: Social support negatively predicts depressive symptoms through the mediating effect of positive coping.

### 1.3. Regulatory Role of Neuroticism

Personality is critical in reducing the risk of mental disorders and is crucial in forming and maintaining healthy interpersonal relationships [41]. Personality traits are persistent automatic patterns of thoughts, feelings, and behaviors that distinguish individuals from one another [42]. Neuroticism is a persistent personality trait and a relatively stable mental structure usually characterized by negative emotional experiences. It is thought to be linked to mood disorders, including depression, and has important public health implications. Individuals with high neuroticism scores tend to be emotionally unstable and more prone to emotional distress and psychological disorders [43]. In contrast, individuals with low neuroticism scores tend to be conceptually well-adjusted, emotionally stable, and able to cope with stressful challenges through positive strategies [41,44]. As a personality factor reflecting individual differences, high neuroticism can explain individual depressive symptoms and sleep quality [45], while low neuroticism can improve life satisfaction [46].

Although social support may indirectly influence depressive symptoms through positive coping, the effect of coping style on depressive symptoms may be moderated by individual factors. For example, self-esteem moderates the relationship between problem-solving strategies and depressive symptoms in college students [47]. Therefore, there is a need to explore factors that can strengthen or weaken the association between social support, positive coping and depressive symptoms. Neuroticism may also be an essential individual differentiator between the two.

Graduate students are at a critical stage of psychological and personality development, and their coping styles and strategies often affect depressive symptoms. Some studies have found that high neuroticism may be an individual factor affecting maladjustment and may increase the risk of depression [48,49]. During the COVID-19 pandemic, highly neurotic college students tended to adopt adverse coping styles [50], which could easily lead to negative emotions and psychological symptoms. This is because people who are highly neurotic are more likely to have psychological distress and irrational thoughts and are less able to control their impulses. In addition, compared with active coping and low neuroticism, high neuroticism, and avoidance coping styles had adverse effects on the mental health of college students during the pandemic [51]. The relationship between personality traits and coping strategies suggests that highly neurotic individuals are at a greater risk of experiencing psychological distress because they often employ negative coping styles, such as avoidance coping. Previous research has focused on the effects of high neural activity and negative coping strategies on mental health. However, how positive coping styles and social support apply to graduate students is not clear. Therefore, the new study examined how this group coped with depressive symptoms during the coronavirus crisis and the moderating role neuroticism played. Therefore, we proposed hypothesis H3: Neuroticism modulates the latter half of the social support pathway, influencing depressive symptoms through positive coping.

In summary, this study constructed a moderating mediation model (Figure 1) to explore the relationship between social support and the depressive symptoms of graduate students during the COVID-19 pandemic and its mechanisms, to enrich the literature on the mechanism of depressive symptoms, and provide empirical support for prevention and intervention.

**Hypothesis** **1.**
*Social support predicts depressive symptoms.*


**Hypothesis** **2.**
*Positive coping mediates the relationship between social support and depressive symptoms.*


**Hypothesis** **3.**
*Neuroticism moderates the relationship between positive coping and depressive symptoms.*


## 2. Materials and Methods

### 2.1. Participants

As the COVID-19 pandemic continues, Chinese universities continue to adopt isolation policies limiting students’ activities on campus. Simultaneously, online teaching methods were implemented to reduce personal contact and maintain social distancing. Therefore, this cross-sectional study conducted an online questionnaire survey using Questionnaire Star, a professional online questionnaire website. 

We used cluster sampling to collect data from 1891 graduate students enrolled in the fall of 2021, including 1494 women (79%) and 397 men (21%). After eliminating 22 invalid questionnaires and 57 individuals who did not complete the questionnaire, 1812 valid questionnaires remained, with an effective recovery rate of 95.82%. There were 1436 women (79.24%) and 376 men (20.75%) aged 20–45 years (*M* = 23.30; *SD* = 2.174). Data were collected anonymously. The data collection period was 1–8 October 2021. All procedures involving human participants were approved by the Ethics Committee of Liaoning Normal University (approval reference LL2021046).

### 2.2. Measurements

#### 2.2.1. Social Support

Social support was measured using the Social Support Rating Scale (SSRS) [52]. The scale includes ten items measuring three dimensions: objective support, subjective support, and support utilization. One of the items, for example, is “How many close friends do you have that you could get support and help from?”. The SSRS generally uses a multi-axis evaluation method. A total score of less than 20 indicates lower levels of social support, 20–30 indicates general social support, and 30–40 indicates satisfactory social support. After testing, the reliability analysis showed that the scale’s internal consistency (alpha) coefficient was 0.726. Confirmatory factor analysis using MPlus 8.4 indicated that the scale had good construct validity: χ^2^/df = 6.070, CFI = 0.963, TLI = 0.947, RMSEA = 0.053, SRMR = 0.035.

#### 2.2.2. Positive Coping

The coping style survey was conducted using the Simplified Coping Style Questionnaire (SCSQ) [53]. It comprises 20 items rated on a 4-point scale (0 = do not accept, 3 = often accept) that measures two dimensions: positive coping and negative coping. The positive coping dimension comprises items 1–12. The questionnaire items included “I strive to see the good side of things” and “releasing myself through work, study, or other activities”. In this study, Cronbach’s α for the coping style questionnaire was 0.761, and 0.819 for the positive coping dimension. According to the original structure of the scale, confirmatory factor analysis was performed on the positive coping dimension using MPlus 8.4, and the fitting indexes were χ^2^/df = 8.750, CFI = 0.925, TLI = 0.901, RMSEA = 0.065, SRMR = 0.039.

#### 2.2.3. Depressive Symptoms

Depressive symptoms were investigated using the Beck Depression Inventory Second Edition Chinese version (Beck Depression Inventory, BDI-II) [54,55]. There are 21 items rated on a 4-point scale ranging from 0 (I do not feel sad) to 3 (I am so sad and distressed that I cannot bear it). One item, for example, is “What is your attitude toward the future?”. The total possible score ranged from 0 to 63. Four degrees of depression severity were identified based on the total score: no or minimal depression (0–13), mild depression (14–19), moderate depression (20–28), and severe depression (29 and above). The internal consistency (alpha) coefficient of the scale was 0.866. Confirmatory factor analysis using MPlus 8.4 showed that the scale had good construct validity: χ^2^/df = 4.938, CFI = 0.914, TLI = 0.903, RMSEA = 0.047, SRMR = 0.035.

#### 2.2.4. Neuroticism

Neuroticism was tested using the adult version of the Eysenck Personality Questionnaire (EPQ-A) [56]. The questionnaire was scored using two responses (0 = no, 1 = yes). There were 88 questions, including 24 on the neuroticism dimension. The α coefficients of the total questionnaire and neuroticism were 0.846 and 0.893, respectively. According to the original structure of the scale, confirmatory factor analysis was performed on the neuroticism dimension using MPlus 8.4, and the fitting indexes were χ^2^/df = 4.823, CFI = 0.915, TLI = 0.906, RMSEA = 0.046, SRMR = 0.036.

### 2.3. Data Statistics and Analysis

The data sets were analyzed using SPSS version 26.0 (IBM SPSS, IBM Corp., Armonk, NY, USA). First, we performed common method bias tests, descriptive statistics, and analyses of the prevalence of depressive symptoms. Second, we assessed correlations between sex, age, social support scores, positive coping scores, neuroticism scores, and depressive symptom scores using Pearson product difference correlations. Finally, the scores for social support, positive coping, depressive symptoms, and neuroticism were converted into standard scores. Models 4 and 7 of the PROCESS plugin explored the mediating and moderating effects of social support and depressive symptoms. The significance level was set for all statistical analyses at *p* < 0.05 (two-sided).

## 3. Results

### 3.1. Common Method Deviation Test

Collecting data using self-report may lead to common method bias. This was controlled for in this study using an anonymous survey and the reverse scoring of some items. Simultaneously, Harman’s single-factor test of common method bias was used. The results showed that there were 14 factors with an eigenvalue greater than 1, of which the cumulative variance explained by the first factor was only 19.868%, which was less than the critical value of 40%, indicating that there was no serious common method bias in this study [57].

### 3.2. Level of Depression, Support, and Neuroticism

The prevalence of depression was estimated using the Beck Depression Inventory-II categorical system [54]. In the current sample, 1623 participants (89.60%) showed no to minimal depression (BDI-II scores from 0 to 13), 117 (6.5%) showed mild depression (BDI-II scores from 14 to 19), 65 (3.6%) showed moderate depression (BDI-II scores from 20 to 28), and seven (0.4%) showed severe depression (BDI-II scores of 29 and above). Thus, 10.40% of the participants showed at least mild depression, according to the BDI-II classification system. There was a significant difference between the groups regarding sex (χ^2^_(2)_ = 262.198, *p* < 0.001).

Social support was estimated using the SSRS scoring method [52]. In the current sample, five participants (0.3%) had a low social support (score less than 20 points), 125 (6.9%) had an average social support (score 20–30 points), and 1682 (92.8%) had a satisfactory social support (score more than 30 points). 

The raw neuroticism scores were converted to T-scores, which estimated the neuroticism level [56] based on the EPQ-A. In the current sample, 217 participants (12%) had typical low neuroticism (below a T score of 38.5), 357 (19.7%) showed proneness (between 38.5 and 43.3), 816 (45%) were intermediate (T scores between 43.3 and 56.7), 123 (6.8%) showed proneness (between 56.7 and 61.5), and 299 (16.5%) showed typical high neuroticism (T score 61. +).

### 3.3. Descriptive Statistics and Correlation Coefficients of Variables

The overall average score of social support for first-year postgraduate students was 42.77 (*SD* = 8.739; women *M* = 43 [*SD* = 0.317], men *M* = 41.92 [*SD* = 10.157]), and the difference was statistically significant (*t* = 2.138, *p* < 0.05). The overall average score for active coping was 25.97 (*SD* = 5.508; women *M* = 26.04 [*SD* = 5.387], men *M* = 25.70 [*SD* = 5.947]). There was no significant difference between males and females (*t* = 1.075, *p* > 0.05). The overall average neuroticism was 9.21 (*SD* = 5.767; women *M* = 9.28 [*SD* = 5.745], men *M* = 8.94 [*SD* = 5.853]), and the difference was not statistically significant (*t* = 1.009, *p* > 0.05). The overall average score for depressive symptoms was 5.36 (*SD* = 5.979; women *M* = 5.19 [*SD* = 5.573], men *M* = 6.01 [*SD* = 7.300], and the difference was statistically significant (*t* = −2.38, *p* < 0.05).

Correlation analyses were performed for sex, age, social support, positive coping, depression, and neuroticism. The results showed that social support positively correlated with positive coping and significantly negatively correlated with depressive symptoms and neuroticism. In contrast, positive coping was significantly negatively correlated with depressive symptoms and neuroticism, and depressive symptoms were significantly positively correlated with neuroticism (see Table 1 for details).

### 3.4. The Relationship between Support and Depressive Symptoms: A Moderated Mediation Model Test

First, to test Hypotheses 1 and 2, all the variables were converted to standard scores. Model 4 of the SPSS macro program PROCESS [58] was used to test the mediating effect of positive coping between social support and depressive symptoms. The results showed that after controlling for age and gender, social support significantly predicted positive coping (a = 0.438, *SE* = 0.021, *p* < 0.001, 95% CI [confidence interval) (0.397, 0.480) (see Table 2). Social support and positive coping were included in the regression equation simultaneously. Social support significantly predicted depressive symptoms (c′ = −0.386, *SE* = 0.023, *p* < 0.001, 95% CI (−0.432, −0.341), and positive coping significantly predicted depressive symptoms (b = −0.159, *SE* = 0.023, *p* < 0.001, 95% CI [−0.204, −0.114]). The bias-corrected percentile bootstrap test showed that positive coping had a significant mediating effect between social support and depressive symptoms, ab = −0.069, BootSE = 0.012, 95% CI (−0.094, −0.046); there was a partial mediation effect, and the proportion of the mediation effect to the total effect was ab/(ab + c′) = 15.164%.

Second, Model 14 of the SPSS macro program PROCESS was used to test the moderating effect of neuroticism. A moderated mediation model test was used to estimate the parameters of the three regression equations. Equation 1 estimated the overall effect of social support on depressive symptoms; Equation 2 estimated the predictive effect of social support on positive coping; Equation 3 estimated the moderating effect of neuroticism on positive coping and depressive symptoms. In each equation, all predictors were standardized, with sex and age controlled for. A moderated mediating effect exists if the model estimates satisfy the following three conditions: (a) in Equation 1, the total effect of social support on depressive symptoms is significant; (b) in Equation 2, the predictive effect of social support on positive coping is significant; (c) in Equation 3, the main effect of active coping on depressive symptoms is significant, and the effect of the interaction term between neuroticism and active coping is significant [58]. In addition, the variance inflation factors of all predictors in this study were no higher than 1.4, indicating no multicollinearity problem.

As shown in Table 2, all three conditions were satisfied. Equation 1 was significant: social support negatively predicted depressive symptoms, satisfying condition (a). Equation 2 was significant: social support positively predicted positive coping and satisfying condition (b). Equation 3 was significant: positive coping negatively predicted depressive symptoms, and the interaction between neuroticism and positive coping was significant, satisfying condition (c).

To elucidate the interaction effect between positive coping and neuroticism, we divided neuroticism into high and low groups by adding or subtracting one standard deviation from the mean, performing a simple slope test, and drawing a simple effect analysis graph (Figure 2). The results showed that for participants with low grouping or low neuroticism, positive coping had no significant negative prediction of depressive symptoms (Bsimple = 0.049, *t* = 1.880, *p* > 0.05). For participants with high grouping or high neuroticism, positive coping was not significant. The negative predictive effect of depressive symptoms was significant (Bsimple = −0.171, *t* = −7.250, *p* < 0.001).

Altogether, the process through which social support affects depressive symptoms through active coping is mediated by neuroticism. For participants with high neuroticism, the indirect effect of social support on depressive symptoms through active coping was significant (*index* = −0.048, BootSE = 0.010, 95% CI [−0.067, −0.028]).

## 4. Discussion

Empirical support exists for the link between social support and depressive symptoms, in which social support negatively predicts depressive symptoms. However, their internal mechanisms remain ambiguous. This study explored the relationship between social support and depressive symptoms in newly admitted graduate students during the COVID-19 pandemic from the theoretical perspective of the stress-buffering model of depressive symptoms. Our findings demonstrate that social support has an impact on depressive symptoms, which is mediated by the use of positive coping mechanisms. On the other hand, it analyzed when the effect was greater, that is, when the second half of the mediation pathway was regulated by neuroticism. High neuroticism was a significant predictor of depressive symptoms among participants. The results of this study have important theoretical and practical significance for scientific prevention and intervention in regulating depressive symptoms among newly graduated students.

### 4.1. Social Support and Depressive Symptoms of Graduate Students

This study found that the level of social support had a significant negative predictive effect on depressive symptoms. We think that this is due to social interactions preventing depressive tendencies through feelings of boredom or loneliness. Nonetheless, they may also be linked to social support when a person experiences personal challenges or academic stress. Other studies have reported similar associations [16], including during the COVID-19 pandemic [30]. The results of this study support the theoretical hypothesis of the stress-buffer model that social support for graduate students can promote positive mental states, reduce the degree of depressive symptoms, and expand the causal mechanism by which social relationships affect physical health. The social lives of graduate students are hit hard by lockdown measures, which prevent social communication. In the face of large-scale asymptomatic nucleic acid testing measures and the requirement of social distancing in the university environment, strengthening on-campus communication, focusing on the establishment of low-risk social activities, and helping students establish and strengthen social support networks can reduce students’ loneliness and promote their mental health [14]. A meta-analysis showed that social support predicted better mental health functioning and was a protective factor against depression [3], particularly among college students. When individuals receive higher levels of social support, they can increase their resilience to stress, reduce the risk of elevated levels of depressive symptoms, and slow the development of trauma-related psychopathology [21,59]. The prevalence of depression was 10.40%, comparable to the rates reported in previous studies [60]. We found significant differences in the depression rates between male and female students, which is consistent with other studies that found higher levels of depression among girls [10]. It is essential to consider the analysis of the gender ratio disparity between men and women in this study with caution. The mental health of graduate students continues to face an unprecedented challenge, in addition to increased online learning, loss of social contact, and changes in daily life due to the continuation of restrictions on access to campus and social-distancing policies.

### 4.2. The Mediating Role of Positive Coping

This study found that positive coping mediates the relationship between social support and depressive symptoms in graduate students; social support reduces the risk of depressive symptoms by increasing positive coping. These results suggested that positive coping is an important mediating mechanism between social support and depressive symptoms. Previous studies have shown that positive coping plays an important role in alleviating depression symptoms in college students [61] and has a central mediating role in reducing depression levels in female college students [62]. During the COVID-19 pandemic, the depression symptoms of graduate students also follow the approach of “macro-level social support → individual positive response → alleviation of adverse symptoms,” that is, through effective social interaction, graduate students expand their community network, cope with the pandemic and adaptation period, and improve their psychological resilience and ability to cope with pressure, to reduce depressive symptoms effectively and actively, staying mentally healthy. Therefore, the mediating role of positive coping validates the stress response model and integrates the social support stress buffer model into the literature on depressive symptoms in the context of COVID-19, revealing how macro-social support factors affect depressive symptoms through positive coping. Undergraduate and graduate students surrounded by a good support system (family and friends) were less affected by mental health problems throughout the pandemic [13]. A study of secondary school and college students during the pandemic found that perceived high levels of emotional support increased the propensity to respond positively. In contrast, perceived low levels of help, lack of reliable friends, and communication problems with family reduced the propensity to respond positively [63]. In contrast, the results showed that avoidance coping more strongly mediates the relationship between post-traumatic stress and depression in younger adults (aged 18–39 years) [64]. These results indicate that college students adopt different coping styles as mechanisms for the correlation between stress and social and mental health. Our findings suggest that interventions aimed at increasing positive (i.e., approach-based) coping behaviors and maintaining social support despite physical distancing barriers may help support the psychosocial health of graduate students during the COVID-19 pandemic. Therefore, university mental health services during the COVID-19 crisis should consider assessing mental health status and stress management to address these psychological issues.

### 4.3. Regulatory Role of Neuroticism

The present study found that neuroticism moderates the latter half of the social support pathway through positive coping that affects depressive symptoms in graduate students. Among them, high neuroticism had a significant indirect effect on graduate students. Generally, individuals with high neuroticism are usually more focused on the negative components of stressful life events, leading to the rapid arousal and continued engulfing of negative emotional experiences, thus increasing the risk of depressive episodes [65]. This study is the first to explore the regulatory mechanism of the mediating chain of “social support, positive coping, and depressive symptoms” from the perspective of neuroticism and integrates the concepts of social support, coping style, and personality into the study of depressive symptoms. This extended the stress buffer model of depressive symptoms to some extent and analyzed the influence of proximal personality factors on depressive symptoms among graduate students. During the pandemic, college students who are highly neurotic are more likely to experience emotional imbalances and tend to use maladaptive emotion-centered coping styles [50,51]. One explanation for this is that people with high levels of neuroticism are more susceptible to psychological distress and irrational thoughts and are less able to control their impulses [66]. The ongoing pandemic and campus closures have had further adverse effects on graduate students with high neurotic levels, limiting their cognitive and emotional adjustment to environmental adaptation and stress coping. Amid concerns about the potential spread of COVID-19 and policy changes, universities must screen and monitor graduate students’ mental health promptly. Simultaneously, preventive mood regulation and early intervention services can also be provided for certain groups of students at higher risk (for example, those with high neuroticism).

## 5. Conclusions

Research concerning college students’ mental health during the pandemic has largely focused on undergraduates; however, the advantage of this study is that it seeks to explore the effects of social support on depressive symptoms among graduate students, as well as analyze the intermediary and regulatory variables. This study found that (1) social support had a significant negative predictive effect on depressive symptoms among first-year graduate students; (2) positive coping mediated the relationship between social support and depressive symptoms among first-year graduate students; and (3) the indirect effect of social support on depressive symptoms was moderated by neuroticism through positive coping. This study highlights that, during the current pandemic, graduate students have limited social support and may experience symptoms of high neuroticism or depression. Longitudinal studies are needed to explore the long-term predictive effects of social support and coping styles on mental health over time. Follow-up studies could analyze the relationship between different types of social support and depressive symptoms in graduate students, particularly the effects of participation in online social activities. As the duration and severity of the outbreak are expected to be longer, future research must develop and implement protective measures to improve the mental health of graduate students, such as online mindfulness and meditation.

## Figures and Tables

**Figure 1 ijerph-20-03431-f001:**
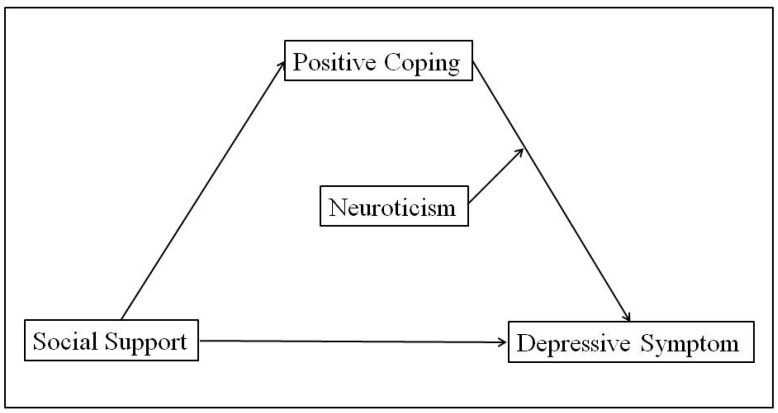
The mediating role of positive coping and the moderating role of neuroticism.

**Figure 2 ijerph-20-03431-f002:**
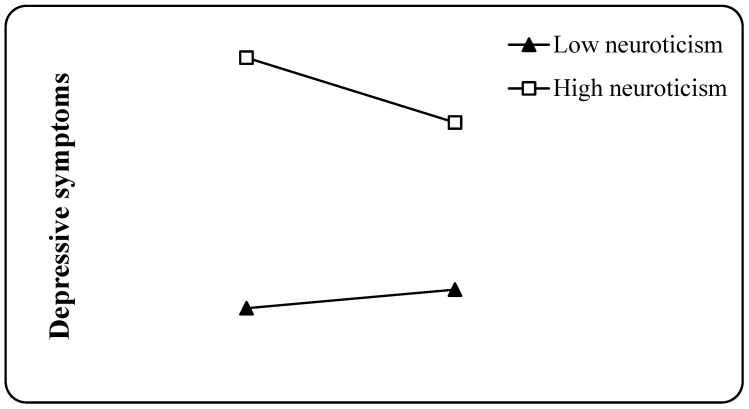
Neuroticism moderates the relationship between social support and postgraduates’ depressive symptoms.

**Table 1 ijerph-20-03431-t001:** Summary of descriptive statistics and correlation coefficients.

	*M* ± *SD*	1	2	3	4	5	6
1. Gender ^a^	1.21 ± 0.406	—					
2. Age	23.30 ± 2.174	0.135 **	—				
3. Social support	42.77 ± 8.739	−0.050 *	0.078 **	—			
4. Positive coping	25.97 ± 5.508	−0.025	0.067 **	0.442 **	—		
5. Depressive symptoms	5.36 ± 5.979	0.056 *	−0.030	−0.458 **	−0.331 **	—	
6. Neuroticism	9.21 ± 5.767	−0.024	−0.104 **	−0.419 **	−0.309 **	0.659 **	—

Note: ^a^ Gender was treated as a dummy variable: 1 = female, 2 = male. * means *p* < 0.05. ** means *p* < 0.01, the same below.

**Table 2 ijerph-20-03431-t002:** The moderated mediating effect test of social support on depressive symptoms.

	Equation 1 (Criterion: Depressive Symptoms)			Equation 2 (Criterion: Positive Coping)			Equation 3 (Criterion: Depressive Symptoms)		
	*β*	*SE*	*t*	*β*	*SE*	*t*	*β*	*SE*	*t*
Age	0.001	0.010	0.082	0.015	0.009	1.583	0.019 *	0.007	2.487
Gender	0.081	0.052	1.552	−0.019	0.052	−0.366	0.122 **	0.041	2.936
Social support	−0.457 ***	0.021	−21.763	0.438 ***	0.021	20.695	−0.196 ***	0.019	−9.924
Positive coping							−0.060 **	0.018	−3.214
Neuroticism							0.549 ***	0.018	29.295
Positive coping * Neuroticism							−0.110 ***	0.016	−6.812
*R* ^2^	0.221			0.196			0.496		
*F*	161.336 ***			147.034 ***			297.160 ***		

Note. * *p* < 0.05, ** *p* < 0.01, *** *p* < 0.001.

## Data Availability

The data for this study are available from the corresponding author upon reasonable request.

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
