# Peer review of "Impact of Social Support on Depressive Symptoms among Postgraduates during the Coronavirus Disease 2019 Pandemic: A Moderated Mediation Analysis"

_ijerph, 2023, doi:10.3390/ijerph20043431_

Round 1

Reviewer 1 Report

1 The introduction needs to be significantly revised. It does not develop a clear justification for undertaking the present study.  

2 The innovation of this article mainly lies in the investigation of depressive symptoms among graduate freshmen during the COVID-19 pandemic, but the article does not elaborates the main existing problems of graduate freshmen during the COVID-19 pandemic. More specifically, why are they more likely to be depressed during the COVID-19 pandemic.

3 What is puzzling me is why graduate freshmen are selected in this study, since many of the problems described in the article are common to all graduate students. Some of the problems may even be worse in students from other grade levels.

4 The discussion needs to be significantly revised in order to demonstrate the relevance and value of the present findings. For example, the author should consider the potential impact of the COVID-19 epidemic on these results.

5 The manuscript should be modified by native English speakers.  

Author Response

Response to Reviewer 1 Comments

Point 1: The introduction needs to be significantly revised. It does not develop a clear justification for undertaking the present study. 

Response 1: Thanks to the expert's opinion, we have revised according to your opinion, rewrote the introduction part, resorted out the overall framework and the logical relationship of sentences, and added new literature. Changes are marked in blue.

Point2: The innovation of this article mainly lies in the investigation of depressive symptoms among graduate freshmen during the COVID-19 pandemic, but the article does not elaborates the main existing problems of graduate freshmen during the COVID-19 pandemic. More specifically, why are they more likely to be depressed during the COVID-19 pandemic.

Response 2: Thanks for the experts' opinions. As you said, if introducing new graduate students, the characteristics of this group should be highlighted. In fact, this article only introduces the graduate student community and does not highlight the freshmen part of it. We have changed the freshmen to graduate students.

Point 3: What is puzzling me is why graduate freshmen are selected in this study, since many of the problems described in the article are common to all graduate students. Some of the problems may even be worse in students from other grade levels.

Response 3: As in the previous revision, we have changed new graduate students to graduate students.

Point 4: The discussion needs to be significantly revised in order to demonstrate the relevance and value of the present findings. For example, the author should consider the potential impact of the COVID-19 epidemic on these results.

Response 4: Thank the reviewers for their comments. The discussion section has been rewritten. Increased the impact of the covid-19 pandemic and increased the relevant research in this paper. All revisions have been marked in blue and references have been recalibrated.

Point 5: The manuscript should be modified by native English speakers. 

Response 5: Thank the reviewers for their comments. We've used Editage to polish the language, along with certificate of editing.

Reviewer 2 Report

This study is very interesting.

However,
to improve the manuscript, I suggest the following modifications, and I have a few questions.

Introduction:

1.       Line 113- 114,123. Please restructure the paragraph about the relationship between macro environment and social support.

2.       Please rename the figure 1 and 2, and give more comprehensive name with clarification of idea of figures.

Materials and methods.

Participant – line 182-192

3.       Please give the explanation about your sampling strategy.

4.       What was the overall amount of newly admitted graduate students in Chinese universities?

5.       What was the overall proportion of female and male of newly admitted graduate students in Chinese universities?

6.       What was the response rate?

Measurements;

7.       Please describe more clear the SCSQ scale, please add the 2 group item characteristics.

Results

8.       Please  describe the level of depression, support and neuroticism in separate sub point, before you provide the data about the correlation variables.

Discussion

9.       Please modify the sentence in line 337-338 for more clearly understanding.

10.   Please add the limitation about gender proportion in your study group.

Conclusions

11.   What future studies do the authors suggest?

Author Response

Response to Reviewer 2 Comments

Point 1: Line 113- 114,123. Please restructure the paragraph about the relationship between macro environment and social support.

Response 1: Thank you for your review comments.

We have reorganized the paragraph on the relationship between the macro environment and social support, and the revisions have been highlighted in blue.

Point 2: Please rename the figure 1 and 2, and give more comprehensive name with clarification of idea of figures.

Response2: Figures 1 and 2 have been renamed and their meanings explained. Changes have been marked in blue.

Point 3: Please give the explanation about your sampling strategy.

Response 3: Cluster sampling method is adopted in this paper, which has been indicated in the paper.

Point 4: What was the overall amount of newly admitted graduate students in Chinese universities?

Response 4: A total of 1891 new graduate students were enrolled. For details, see section 2.1 Participants.

Point 5: What was the overall proportion of female and male of newly admitted graduate students in Chinese universities?

Response 5: We used cluster sampling to collect data from 1891 graduate students enrolled in the fall of 2021, including 1494 women (79%) and 397 men (21%). For details, see section 2.1 Participants.

Point 6:  What was the response rate?

Response 6: A total of 1812 effective questionnaires were collected, with effective recovery of 95.82%.

Point 7: Please describe more clear the SCSQ scale, please add the 2 group item characteristics.

Response 7: To add descriptions of the characteristics of the SCSQ scale of the two groups, see 2.2.2. Positive coping

Point 8: Please  describe the level of depression, support and neuroticism in separate sub point, before you provide the data about the correlation variables.

Response 8: Levels of depression, support, and neuroticism have been described separately, see 3.2. Level of depression, support, and neuroticism for details.

Point 9: Please modify the sentence in line 337-338 for more clearly understanding.

Response 9: The sentence has been revised. For details, see line 372-373 of the revised version.

Point 10: Please add the limitation about gender proportion in your study group.

Response 10: Restrictions on the ratio of men to women have been added, see lines 402-404 for details.

Point 11: What future studies do the authors suggest?

Response 11: In terms of theory, it is imperative to examine the long-term influence of social support and coping styles on mental health, while in terms of practice, it is essential to enhance the intervention of graduate students' mental health during the pandemic. For details, see lines 472-481.

Reviewer 3 Report

This is an article focused on an area with good justification and the need to deepen the topic "The relationship between graduate freshmen social support and depressive symptoms during the COVID-19 pandemic: Moderate mediating effect of positive coping and neuroticism" In general, most of the methods have been used to a good standard and have been well described. I have some comments and suggestions to help improve clarity in parts of the document:

In the Title: It would be interesting to include the type of study.

In the Summary: The summary is very brief and more relevant information should be included:- It would be interesting to include a clearly defined objective of the study... The objective of the study is... to help a better reading and understanding...

• The duration of the study.

• The type of study.

• It is necessary to deepen the methodology and results.

In the introduction:

I recommend reworking the introduction section and making a stronger theoretical framework for your study. In my opinion, the justification of the main objective is not clear. Please specify them and connect them to the literature review.

Methods:

• I think it would be more interesting to organize the first section of material and methods in a more structured way and include sections such as: study design, participants (include inclusion and exclusion criteria, participant flowchart), ethical considerations

• Include how the participants were selected, specify if it was by random sampling or not.

In the Conclusion:

• Does it respond to the main objective of the study? In my opinion, the conclusion is very brief and should answer the main objective of the study.

• It would be interesting to also include the strengths of this study.

Bibliography:

Replace old references with newer ones. For example, S Cohen, T A Wills (1985)

Author Response

Response to Reviewer 3 Comments

Point 1: In the Title: It would be interesting to include the type of study.

Response 1: Thank you for your review.

The title has been amended to include the type of study.

Point 2: In the Summary: The summary is very brief and more relevant information should be included:- It would be interesting to include a clearly defined objective of the study... The objective of the study is... to help a better reading and understanding...

  • The duration of the study.
  • The type of study.
  • It is necessary to deepen the methodology and results.

Response 2: The abstract has been revised to include an introduction to the duration of the study, the type of study, and a deeper section on methods and findings.

Point 3: I recommend reworking the introduction section and making a stronger theoretical framework for your study. In my opinion, the justification of the main objective is not clear. Please specify them and connect them to the literature review.

Response3: The introduction has been rewritten. Added to the introduction of the background, focus on the analysis of the relationship between research variables and theoretical support. The logical relationship of each part of the introduction is rearranged to make the sentence more accessible. The changes have been highlighted in blue.

Point 4: I think it would be more interesting to organize the first section of material and methods in a more structured way and include sections such as: study design, participants (include inclusion and exclusion criteria, participant flowchart), ethical considerations

Response 4: This part of the revision adds sampling method, exclusion method and ethical approval.

Point 5: Include how the participants were selected, specify if it was by random sampling or not.

Response 5: All participants were selected by cluster sampling method.

Point 6: Does it respond to the main objective of the study? In my opinion, the conclusion is very brief and should answer the main objective of the study. It would be interesting to also include the strengths of this study.

Response6: The analysis of research objectives and strengths is added to the conclusion. For details, see the discussion section, which has been highlighted in blue.

Point 7: Bibliography: Replace old references with newer ones. For example, S Cohen, T A Wills (1985)

Response 7: The reference format has been re-edited to the new format.
